# Hydrogen Embrittlement and Oxide Layer Effect in the Cathodically Charged Zircaloy-2

**DOI:** 10.3390/ma13081913

**Published:** 2020-04-18

**Authors:** Grzegorz Gajowiec, Michał Bartmański, Beata Majkowska-Marzec, Andrzej Zieliński, Bartosz Chmiela, Marek Derezulko

**Affiliations:** 1Department of Materials Engineering and Bonding, Gdansk University of Technology, Narutowicza str. 11/12, 80–233 Gdansk, Poland; grzegorz.gajowiec@pg.edu.pl (G.G.); michal.bartmanski@pg.edu.pl (M.B.); beamajko@pg.edu.pl (B.M.-M.); mderezulko@gmail.com (M.D.); 2Department of Advanced Materials and Technologies, Silesian University of Technology, Krasińskiego 8, 40–019 Katowice, Poland; Bartosz.Chmiela@polsl.pl

**Keywords:** oxide layers, hydrogen embrittlement, hydrogen uptake, hydrogen diffusion, zirconium alloys, mechanical properties, nanoindentation, metal softening, metal hardening

## Abstract

The present paper is aimed at determining the less investigated effects of hydrogen uptake on the microstructure and the mechanical behavior of the oxidized Zircaloy-2 alloy. The specimens were oxidized and charged with hydrogen. The different oxidation temperatures and cathodic current densities were applied. The scanning electron microscopy, X-ray electron diffraction spectroscopy, hydrogen absorption assessment, tensile, and nanoindentation tests were performed. At low oxidation temperatures, an appearance of numerous hydrides and cracks, and a slight change of mechanical properties were noticed. At high-temperature oxidation, the oxide layer prevented the hydrogen deterioration of the alloy. For nonoxidized samples, charged at different current density, nanoindentation tests showed that both hardness and Young’s modulus revealed the minims at specific current value and the stepwise decrease in hardness during hydrogen desorption. The obtained results are explained by the barrier effect of the oxide layer against hydrogen uptake, softening due to the interaction of hydrogen and dislocations nucleated by indentation test, and hardening caused by the decomposition of hydrides. The last phenomena may appear together and result in hydrogen embrittlement in forms of simultaneous hydrogen-enhanced localized plasticity and delayed hydride cracking.

## 1. Introduction

Some Zr alloys are used in the nuclear industry for fuel claddings [1,2,3], reflectors in light water reactors [4], and in spent nuclear fuel reprocessing plants [5,6]. Zirconium alloys are applied to manufacture nuclear fuel pellets due to their low thermal neutron capture cross-section, proper strength properties, and excellent corrosion resistance in the cooling medium [7]. The nuclear fuel pellets are made of the Zr-Sn Zircaloys, Zr-Nb E110, E125, and E635, Zr-Sn-Nb Zirlo, Zr-Nb M5, and X5A alloys [7,8,9,10], Zr-Nb and Zr-Nb-Fe [11], and the 702 alloys [12,13]. 

The most important degradation mechanisms of zirconium alloys in the nuclear industry comprise high-temperature oxidation, delayed hydride cracking, electrochemical corrosion of waterside of fuel pellets, and pipelines, and creep at elevated temperatures [2,11]. The most severe nuclear event, the loss-of-coolant-accident (LOCA), goes through high-temperature oxidation of fuel pellets followed by rapid quenching because of an emergency coolant reflooding into the reactor core. As a consequence, the fuel rod integrity disappears at high temperature, at first by a ductile failure at the beginning, and then by the brittle failure after high-temperature oxidation in steam [1,14,15,16,17,18,19,20]. The leakage of the cooling medium may result in overheating of the reactor to 1000–1200 °C, which is followed by decomposition of water vapor to oxygen and hydrogen, and finally, by the reaction of vapor, oxygen, and hydrogen with zirconium fuel pellets [21]. 

The oxidation of zirconium alloys results in the formation of oxide layers, whose chemical and phase composition, microstructure, thickness, adhesion to the zirconium substrate, and permeability to hydrogen, significantly depend on temperature, environment, oxidation time, and alloy [2]. The structure and thickness of oxides and external temperatures are essential for hydrogen uptake by the metal, which can proceed by either the movement of hydrogen atoms (protons) to the bare metal through cracks and crevices, or by diffusion of protons through the oxide layer. The monoclinic ZrO_2_ appears at moderate temperatures and as tetragonal oxide between 1170 °C and 2370 °C [22,23]. However, the transformation of monoclinic into a tetragonal form may occur at low and moderate temperatures under thermal stresses [22]. This phase transformation results in breakaway oxidation followed by the descaling of the oxide layers, presumably caused by either different volume expansion coefficients of matrix and oxide phase, by the presence of two various crystalline oxide structures, or by oxide porosity [24]. The breakaway oxidation usually starts at about 950–1200 °C after a particular initiation time ranging from 2 to 12 ks [24,25]. The characteristics of the oxide layer and temperature are the main factors determining the hydrogen rate uptake. 

Hydrogen appears in the zirconium matrix as an interstitial atom that can be bound by crystal imperfections and in the form of hydrides. Its interstitial solubility drastically decreases with decreasing temperature. According to Suman et al. [26], the limit hydrogen solubility in zirconium changes with temperature from 50% at. at 1000–1300 K to 0.7% at. at 573 K, and only 10^−4^% at. at room temperature. In other research [27], slightly different values were reported: 5.6% at. at 500 °C, 1.5% at. at 300 °C, and 0.01% at. at room temperature. The maximum hydrogen content of H/Zr = 0.9 (which is still in the single-phase β-Zr region) was obtained at 1230 K and 100 kPa of gas hydrogen pressure [18]. 

The diffusivity of H in Zr can be expressed as 5.468 × 10^−7^ exp(−45293/RT) m^2^/s [28]. Two stable hydride phases, δ-hydride (fcc) and ε-hydride (fct), and a metastable phase, γ-hydride (fct), may appear at different temperature and hydrogen content [29]. The δ-hydride phase was predominant at hydrogen content up to 1250 wt. ppm, while at about 3000 wt. ppm of hydrogen, a significant amount of γ-hydride was observed, and at the highest hydrogen content, over 6000 wt. ppm, the ε-hydride phase appeared [30]. The hydride phase may occur after in situ (gas) or ex situ (cathodic) exposure of the Zr surface to hydrogen even at 20 °C [31], decomposing with temperature increasing to 600 °C. 

The hydrogen solubility in the oxide layer is about 10^−3^ to 10^−4^ mol H/mol oxide and decreases with increasing temperature [32]. The hydrogen uptake in the initial phase is controlled by hydrogen diffusion through the growing oxide layer, and it is proportional to the hydrogen absorption time, t, as of t^3/8^ [21]. An initially high hydrogen absorption rate may decrease with increasing oxide thickness up to the oxide phase transition region [2]. The oxide pores provide short circuits for oxygen and hydrogen diffusion [33], as well as the sites to form molecular hydrogen [34]. A presence of oxygen may affect hydrogen absorption, as hydrogen can be trapped by oxygen atoms positioned along the preferred path creating the O-H bonds [28,35]. 

The presence of hydrogen has diverse effects on mechanical properties, which are related to hydrogen content and temperature. The softening is assumed to be due to interstitial hydrogen solution and appears at low hydrogen content resulting in a decrease in Young’s modulus, shear modulus, and microhardness [36,37], and demonstrates an entirely ductile fracture [20,38]. The presence of hydrides at higher hydrogen content usually manifests in often slightly increased, and less often decreased room tensile strength, decreased ductility, and elongation [39,40,41,42]. According to [43], the more significant effect of hydrogen content on mechanical characteristics was noticed at temperatures between 20–170 °C.

In nanoindentation studies, the results were divergent. Setoyama et al. [44] observed that H absorption resulted in either hardness or softening, depending on H content. On the contrary, Ito et al. [45] reported no difference in mechanical properties for the Zr alloy with or without hydrogen. According to Kuroda et al. [46], the zirconium hydride possessed a higher nano-hardness, 4.3 GPa, than that of zirconium, 2.4 GPa. The Young’s modulus was reported as about 95 GPa [47] for hydrogen contents of 150 and 1200 ppm and 83 GPa for the zirconium matrix. At room temperature, Young’s modulus values were assessed as of 99 GPa and 133 GPa, and the hardness values as of 2 GPa and 4.6 GPa for the Zircaloy-4 metal matrix and δ-hydride precipitate, respectively [48]. At room temperature, Suman et al. found [49], that for Zircaloy-4 alloy, the hardness, yield strength and ultimate tensile strength for the matrix were 2 GPa, 310 MPa, and 436 MPA, respectively, and for the δ-hydride 4.7 GPa, 707 MPa, and 998 MPa, respectively. 

The fracture behavior is sensitive to the presence of hydrides. The solid solution of hydrogen resulted in a ductile fracture with dimples associated with metallic ligaments [50,51,52]. At room temperature, the brittle behavior was observed at room temperature at 220 wt. ppm of hydrogen [52]. The cleavage facets appeared on the fracture surfaces of zirconium, together with some dimples in Zr-Mn alloys [53]. However, in another research, except for some specimens with high H concentration at room temperature, the macroscopic fracture behavior was ductile [38]. At the grain size below 30 μm, the cracks were small and independent, while above 500 μm of grain size, they became more abundant [54]. 

Despite a significant number of investigations on hydrogen embrittlement of zirconium alloys, two problems have been the objects of only several studies. The first open question is how the oxidation at different temperatures may affect the high fugacity hydrogen uptake by the bare alloy, as determined by the oxide layer thickness, microstructure, and hydrogen diffusion paths. The next question regards the mechanism and contribution to the hydrogen embrittlement of the interstitial hydrogen atoms (mobile or pinned, either by weak or strong traps) and of the hydride precipitates, with both forms appearing together at room temperature. This research is aimed at making further progress in this area by characterizing the following: (i) the effects of oxide layers, formed at different temperature in air, on hydrogen uptake, the appearance of hydrides and cracks, and change in tensile properties, and (ii) the influence of hydrogen content on mechanical and nanomechanical properties within a surface layer, beginning from the low to high hydrogen content values at room temperature during hydrogen absorption and desorption. 

## 2. Materials and Methods 

The Zircaloy-2 alloy (Sn 1.36, Fe 0.18, Cr 0.11, Fe 0.07, C 0.013, Si 0.091, Al 0.023, others < 0.01 wt. pct., Zr the remainder) delivered by AREVA (AREVA, Warszawa, Poland) was tested. The specimens were cut from the tubes (10.5/12 mm of diameter) into slices with a thickness of 4 mm (for microscopic examinations of oxidized and hydrogen-charged specimens), or from the rods of 11 mm in diameter (for tensile strain tests, LECO assessment, and nanoindentation tests). The samples were polished with abrasive papers, No. 2000 as the last, and diamond paste, 3 μm of granulation (the Struers, Denmark, provided all sandpapers and diamond pastes) to the 0.01 µm of roughness parameter Sa, and then cleaned in deionized water in an ultrasonic bath and dried in warm air. 

The experimental plan was comprised of several stages, as shown in Table 1. In the first stage (No. 1), the effects of oxidation temperature on the oxide layer thickness and degradation were analyzed, establishing in such a way the proper oxidation temperatures for the next tests. Afterward, the slow strain rate tests, together with or after hydrogen loading, were performed to select the procedure for an assessment of mechanical properties (No. 2a). For some specimens oxidized at high temperature, the cathodic charging at 300 mA/cm^2^ was also done to know whether the significant increase in current density could result in any effects in slow strain rate tests (performed simultaneously with hydrogen charging). In the next steps, the effects of the oxide layer thickness on the hydrogen content (No. 2b) and the presence of hydrides in bulk (No. 2c), were carried out determining the barrier function of the oxide layer. In the final steps, the effects of hydrogen amount, determined by cathodic current density (No. 3), and hydrogen desorption, determined by desorption time (No. 4), on the nanoindentation mechanical properties of nonoxidized specimens were examined. All tests were performed on at least three specimens for any process determinant, and were randomly selected among all those prepared at the beginning of the research. 

The oxidation tests were performed in the electric oven (KO14 Typ, VEB Elektro Industrieofenbau Römhild; now the ELIOG Industrieofenbau GmbH, Römhild, Germany). The specimens were put into the already heated oven, and taken outside immediately after the oxidation termination. 

After oxidation, the samples were hydrogen charged at room temperature in a 1 M H_2_SO_4_ solution, neither stirred nor deaerated. The cathodic hydrogen charging was applied, its parameters selected based on some previous reports [20,55,56,57,58]. The charging time was 72 h, the cathodic current density 80 mA/cm^2^, and the imposed voltage was varied from 2 to 30 V, related to the electrical resistance of oxide layers and in an attempt to maintain the fixed current value. Afterward, the specimens were removed from the solution and cleaned with deionized water. The samples were then annealed at 400 °C for 4 h in air, following earlier research [20,42,50,57,59], a time sufficient to obtain the homogenous bulk distribution of hydrogen across the whole specimen. 

All hydrogen-charged specimens, used for slow strain rate tests (SSRT) and microscopic examinations of cross-sections were prepared as the round samples cut from the rods in the direction transverse to the rolling direction, to the dimensions of 120 mm in total length, 50 mm in length of the working part, and 4 mm in diameter. The SSRT tests were performed with the self-designed and constructed tensile machine at a strain rate 10^−5^ s^−1^ in laboratory air. The samples were either (i) simultaneously charged and tensed or (ii) hydrogen-charged, then heated at 400 °C for 4 h and finally tensed at the above strain rate till the fracture. The tensed specimen constituted a cathode, and the platinum was an anode. The force and elongation were monitored during the tensile tests, and the fracture faces examined after the test. Three tests were made for each specimen.

The microscopic examinations were performed with the JEOL 7800 F (JEOL, Tokyo, Japan) scanning electron microscope on the cross-sections of polished samples before any test (reference specimens), after oxidation and after hydrogen charging. The etching of cross-sections was made with the solution containing 45 cm^3^ of 65% HNO_3_, 10 cm^3^ of 40% HF and 45 cm^3^ of deionized water (all reagents delivered by the Polskie Odczynniki Chemiczne, Gliwice, Poland). The thickness of oxide layers was calculated for each specimen as a mean value of five independent measurements, made at an equal distance, close to the oxide thickness. After tensile tests, the microscopic examinations were performed on fracture faces.

The nanoindentation tests were made immediately after the termination of the cathodic charging (at maximum hydrogen uptake). For the highest current density, similar measurements were performed at a different time after charging termination (during hydrogen desorption). The nanoindentation measurements were performed with the NanoTest™ Vantage (Micro Materials) equipment using a Berkovich three-sided pyramidal diamond. Twenty-four independent measurements were made for each sample in its different areas. The maximum applied force was equal to 50 mN, the loading and unloading time were set up at 10 s and the dwell period at maximum load was 5 s. The distances between the subsequent indents were 50 μm. During the indent, the load–displacement curve was determined by the Oliver and Pharr method. From the load–penetration curves, surface hardness (H), reduced Young’s modulus (E), plastic and elastic work were calculated using the integrated software. 

The hydrogen content was measured using the LECO ONH836 elemental analyzer (LECO, U.S.A.) by the inert gas fusion technique at the Silesian University of Technology. The hydrogen from the sample was converted into a H_2_O and measured by the nondispersive infrared absorption technique. Each value was a mean value based on six independent measurements.

## 3. Results

### 3.1. Surface Topography

Figure 1 shows that the surfaces of specimens oxidized in a range of temperatures from 500 °C to 1000 °C were distinctly affected by heat treatment. The surfaces after oxidation at 400 °C and 300 °C not shown here, were perfectly smooth and homogenous as no oxide layer was formed. The deterioration of the surface started already at oxidation at 500 °C. Some short cracks appeared on surfaces oxidized at 700 °C, and they propagated and resulted in the descaling of the oxide layers obtained at 900 °C and 1000 °C. 

### 3.2. Microstructure and Thickness of Oxide Layers

Figure 2 illustrates that the oxide layers are well adjacent to the substrate if the specimens are oxidized at temperatures up to 700 °C. The layers were slightly waived and showed small submicron pores between the elongated columnar grains. There were only single shallow cracks that did not extend across the entire oxide layer. The oxidation at 900 °C and 1000 °C resulted in numerous transverse (circumferential) and single longitudinal cracks passing through the whole oxide layer. Several groups of such cracks were observed with a width of up to 1 μm. The pile-ups of the cracks resulted in the descaling of the oxide layers. 

Figure 3 shows the distribution of primary elements within the oxide layer. The oxidation in the air caused the appearance of zirconium, oxygen, and, at some distance, also nitrogen.

Figure 4 illustrates the relation of oxide layer thickness to temperature. The layer thickness changed from some tens or a hundred nanometers for nonoxidized or oxidized alloys at temperatures of 350–500 °C. At higher temperatures, the growth followed kinetics laws. The width of the oxide layer was the largest near the cracks. The previously performed monitoring of oxidation time [60] showed that after oxidation at 900 °C, the cracks appeared on the surface already after 20 min of oxidation and were relatively long, even up to 35 μm. The oxide layer was substantially thicker in this area. The monotonic increase in the oxide layer thickness is in full accordance with previous results.

### 3.3. Hydrogen Permeation

The hydrogen charging tests were conducted for all specimens at cathodic current density of 80 mA/cm^2^, and exceptionally also at 300 mA/cm^2^. For samples oxidized at 350 °C, i.e., for those applied in slow strain rate tests, the voltage value necessary to keep the current constant was stable, at about 2.6 V. 

However, after oxidation at 700 °C at such low voltage value, the current became very low because of the resistivity of the thick oxide layer. Therefore, the voltage was raised to 10 V (higher voltage resulted in very serious bubbling). Even at such voltage value, the current density was below the expected value, as shown in Figure 5. For specimens oxidized at 900 °C, the current dropped down to a value of zero, even at 30 V of the voltage value. These tests demonstrate the effect of oxidation temperature on cathodic charging parameters, current and voltage, resulting in the significant increasing electrical resistance of the oxide layer. Neither dissolution nor damage of oxide layers was observed at any test, so the oxide layer became an effective barrier.

### 3.4. Hydrogen Uptake 

The results of hydrogen content in samples as-received, nonoxidized and oxidized at different temperature for 30 min, hydrogen charged at cathodic current density 80 mA/cm^2^ and 72 h charging time and then subject to homogenizing heat treatment at 400 °C for 4 h, are shown as the means and standard deviations (SDs) in Figure 6. All average hydrogen contents were relatively low. The highest values were observed for nonoxidized and oxidized specimens at 300 °C. The hydrogen decrease with increasing oxidation temperature can be noticed.

### 3.5. Microstructure of Hydrogen-Charged Specimens as Related to Oxidation Temperature

Figure 7 shows the cross-sections of neither oxidized nor hydrogen charged reference samples, and hydrogen-charged samples without oxidation, or after oxidation at different temperatures. The lines of hydrides about 100 μm in length, appear randomly distributed in the matrix and branched. Still, they are absent in specimens oxidized at temperatures of 500 °C, 600 °C and above (not shown here).

The fracture faces of some selected specimens oxidized at different temperatures and hydrogen charged are shown in Figure 8. For specimens simultaneously hydrogen-charged and tensed shown in Figure 8a, and for reference specimens shown in Figure 8h, a similar ductile fracture is noticed because of minimal hydrogen mobility at room temperature, and its presence only within the thin subsurface layer. On the contrary, the hydrogen charging followed by homogenization at 400 °C for 4 h, a time long enough to achieve the uniform hydrogen concentration across the sample, resulted in deep and frequent internal cracks caused by the decomposition of hydrides. When the oxidation temperature was raised to 700 °C, no hydride-related cracks were observed, as shown in Figure 8c–e. The fracture morphology changed at high-temperature oxidation due to the phase transformation of zirconium in this area of temperatures, as shown in Figure 8d–g. Regarding the mechanism and pathways of cracks as shown in Figure 8b, the cracks were predominantly transgranular, the fracture itself was quasi-cleavage, and large areas of totally plastic deformation surrounded the cracks.

### 3.6. Microstructure of Hydrogen-Charged Alloy as Related to Cathodic Charging Density

Figure 9 shows that in the absence of the artificial oxide layer, hydrogen can enter the alloy and form lines of hydrides after the homogenizing heat treatment. They appeared even at a cathodic current density of as low as 40 mA/cm^2^, in an amount increasing with applied current. Figure 10 confirms this observation. The fracture mechanism was mostly ductile or mixed, and to a great extent, brittle near the cracks. 

### 3.7. Mechanical Properties in Tensile Tests

The stress–strain curves are shown in Figure 11 for specimens nonoxidized (reference), and for those oxidized and hydrogen charged at different conditions. The hydrogen charging had a small and complex effect on tensile properties. The samples, tensed and charged simultaneously, demonstrated the highest tensile strength and those charged, annealed and tensed—a little smaller tensile strength and similar elongation. The noncharged specimens had comparable tensile strength and lower elongation. The oxidation at 900 °C caused an apparent decrease in plasticity. Hydrogen presence slightly increased the elongation of the tested alloys and did not affect the tensile strength within the limits of experimental error.

### 3.8. Nanoindentation Tests During Hydrogen Charging (Hydrogen Uptake)

Figure 12 shows typical stress–strain curves for a single run and a series of nanoindentation tests. The obtained results are shown in Figure 13 and in Table 2. The load–displacement plots were related to hydrogen content. The minima in hardness reduced Young’s modulus, plastic wok, and indent depth were observed. 

The means and standard deviations based on ten independent measurements are shown.

### 3.9. Nanoindentation Tests During Hydrogen Desorption

The results of the planned tests (see Table 1) are shown in Figure 14 and Table 3. The hardness was shown to slowly and monotonously decrease during outgassing, and Young’s modulus possessed a relatively stable value. The plastic work and indent depth slightly increased, and elastic work remained constant.

The means and standard deviations based on ten independent measurements are shown.

## 4. Discussion

### 4.1. Characteristics of Oxide Layers 

The hydrogen embrittlement can manifest itself in a different form and proceed according to various mechanisms. However, the necessary condition for that is always a hydrogen uptake in sufficient amounts, which depends on many factors. Among them, the thickness, chemical and phase composition, morphology, and homogeneity of the oxide layer, can be considered as the crucial intrinsic determinants of the hydrogen absorption rate.

All these factors depend on the oxidation temperature and time, composition and fugacity of a medium, and chemical and phase composition of an alloy [23]. The obtained results here correspond to the previous reports regarding the temperature dependence of the oxide layer growth [25,61,62,63,64,65,66,67], the complex multilayer and multiphase structure [68,69,70,71], growth of the oxide layer in columnar grains [68,72], and the appearance of porosity [25,33,73,74,75] and cracks [76,77,78,79]. Interestingly, it was also noted that a severe descaling of the oxide layer appeared in these investigations at a relatively high temperature. The reason can be the surface characteristics; as we have observed previously, the Zircaloy-2 tubes in the as-received condition showed more advanced cracking and substantial defragmentation at lower temperatures of 700 °C and even 500 °C [80], differentiating them from the polished specimens investigated here, as shown in Figure 1. This factor, the roughness of the alloy surface, seems essential, but is seldom taken into account. For example, the polishing with sandpapers was sometimes applied when testing the oxidation effects of the Zr alloys [24,61,69], and in other studies, the rough surfaces after machining were left [21,34,40,65,81,82].

The thickness of the oxide layers distinctly depends exponentially on the oxidation temperature, as shown in Figure 2 and Figure 4. The complex structure and the presence of both oxygen and nitrogen can be observed in Figure 3, the latter element postulated in the past as essential for degradation of Zr alloys [65]. 

### 4.2. Barrier Effect of the Oxide Layer

The absorbed hydrogen intake was assumed as inversely proportional to the oxidation kinetics [83], except for a nonmonotonic change in the transition region [26]. Despite this knowledge, the influence of the oxide layer characteristics on hydrogen absorption was not often investigated. The applied cathodic charging in cold or heated electrolytes was already utilized [14,20,38,50,56,57,60,84,85,86,87], resulting in the appearance of hydrogen within a thin subsurface layer, mainly as the hydrides and saturated or oversaturated hydrogen interstitial solution. The further annealing (here called homogenizing heat treatment) resulted in a uniform hydrogen concentration. However, the effects of oxide layers on hydrogen absorption by Zr alloys at room temperature, when using high fugacity hydrogen, have never been investigated. The cathodic charging procedure of Zr alloys is significantly different from gas hydrogen charging at elevated temperatures as: (i) at room temperature, hydrogen diffuses very slowly and remains within the oxide or subsurface matrix layer, and (ii) at electrolytic hydrogen charging, the fugacity, being an equivalent of gas pressure, is very high and, simultaneously, (iii) hydrogen solubility and diffusion are very low. Therefore, two aspects should be considered when discussing these results: (i) whether the diffusion of hydrogen through the oxide layer can effectively occur at room temperature, (ii) whether the degradation of the oxide layer is a significant factor of hydrogen absorption.

The diffusion of hydrogen (more strictly, deuterium) through the oxide layers was already shown [34,88,89,90,91,92], but only at elevated temperatures. The important hydrogen barrier effect of an oxide layer thickness of 20 μm around the hydrides was recently postulated [81]. Some of our observations shown in Figure 5 confirm that such a phenomenon can also occur at room temperature. For nonoxidized specimens and those oxidized in range of 350–500 °C, the current density remained constant at the voltage value set at the beginning of the test. The presence of the oxide layer after the oxidation at 700 °C caused the apparent decrease in the current value at the previously set voltage value and, therefore, the increase in voltage became necessary. On the other hand, after oxidation at 900 °C, no current flow was observed during the 72 h long test, even at the high voltage value. It may be assumed that during 3-day charging, hydrogen can diffuse through an approximately 4 µm thick oxide layer up to the bare metal, but it cannot penetrate the layer of about 14 µm thickness. Fick’s equation demonstrates the relation between main diffusion variables as follows:D = x^2^ / t(1)
where D is the hydrogen diffusion coefficient, x the thickness of the oxide layer, and t the charging time. Following that, the hydrogen diffusion coefficient within the oxide structure may be calculated as ranged between 0.6 × 10^−10^ cm^2^/s and 6 × 10^−16^ cm^2^/s. These values correspond to the hydrogen diffusivity of about 10^−14^ cm^2^/s at temperatures of 573–673 K [88,89,90]. These observations suggest that even at room temperature, the hydrogen diffusion through the oxide layer proceeds, and the oxide layers cannot be an effective barrier against hydrogen uptake if the charging time is long enough. On the other side, the interstitial hydrogen solubility is very low at room temperature, and even if hydrogen may slowly migrate, its low solubility at room temperature results in an appearance of the platelets of hydrides at the interface oxide—Zr matrix. They can efficiently oppose further hydrogen migration at room temperature. 

In regard to the second question, it may be expected that a substantial degradation of the oxide layers after oxidation at 900 °C and 1000 ° C should open the natural diffusion pathways, as shown in Figure 1. However, these oxide layers are still practical barriers against hydrogen absorption after 3-day charging, as shown by an absence of hydrides, displayed in Figure 7. It may be attributed to the appearance of a complex, two-zone structure of the oxide layer, as shown in Figure 2. The outer sublayer seems easily damaged, and the inner sublayer stays still well-adjacent to the substrate. This corresponds to the previous research [71] in which the oxide layer has been shown to be composed of two sublayers: the external one extremely permeable to hydrogen even at over 300 °C, and the inner one, much more impermeable to hydrogen. 

The high hydrogen fugacity and homogenizing heat treatment resulted in relatively low average hydrogen contents, useful for the examinations of both hydrogen forms, hydrides, and oversaturated hydrogen interstitial solution in the Zr matrix. It is of note that the hydrogen amount in each test has been sufficient to form hydrides. The barrier effect of the oxide layer is clearly shown after the homogenizing treatment. The hydrides were observed in hydrogen-charged nonoxidized specimens and those subject to oxidation at temperatures of 300 °C, 350 °C and 400 °C, as shown in Figure 6, Figure 8b and Figure 10. The oxide layers formed after oxidation at slightly higher temperatures with a thickness of a few or tens micrometers were then effective barriers, but only during 72 h of hydrogenation. Still, it may not be valid during long-term exposure and a high fugacity hydrogen source.

### 4.3. Effect of Hydrogen on Microstructure

Hydrogen embrittlement (or, strictly speaking, hydrogen-enhanced degradation) may manifest in different forms. Here, two such degradation forms were investigated: (i) irreversible change in the alloy microstructure and (ii) change in mechanical properties.

The first form was examined after hydrogen charging and homogenizing treatment. Heat treatment time and temperature were sufficient for hydrogen migration across the whole sample and likely allowed the achievement of thermodynamic equilibrium between hydrogen in the interstices, that bond by dislocations and pile-ups, or intermetallic phases and hydrogen precipitating as hydrides. During annealing at 400 °C for 4 h, the approximate values of D_H_ were between 10^−5^ to 10^−8^ cm^2^/s [28] and based on the Fick’s equation, the distance of hydrogen migration may be assessed as 0.509 mm to 1.61 cm. The microscopic examinations showed, however, that the distribution of hydrides was regular for both tubular and flat samples and on the whole cross-section area, and thus the applied annealing conditions were adequate for hydrogen to move across the sample. Therefore, the hydrogen diffusion coefficient seems closer instead to the higher of the above values.

The hydrogen solubility at 400 °C can be assessed as about 200 ppm [26,27], far above the hydrogen contents measured here, as shown in Figure 6. During heat treatment, the thin surface platelets of hydrides were dissolved and hydrogen moved into the bulk, but in part, also desorbed from the specimen. When the sample was cooled to room temperature, the hydrides uniformly dispersed precipitated again, as shown in Figure 7 and Figure 9. The results are consistent with previous reports on tests made on hydrogen-charged specimens, homogenized by the annealing at 400 °C, and examined at room temperature [20,38,50,56]. 

The appearance of hydrides is necessary, but not the only condition for delayed hydride cracking (DHC) to occur. DHC is a repeated process that involves hydrogen diffusion in the area ahead of the crack, precipitation of hydride platelets, and fracture in the brittle hydride region [93]. Many factors determine the hydride-induced degradation [94], such as the toughness of hydrides, size, and distribution (parallel or perpendicular to the rolling direction) of hydrides, hydrogen content, the value of tensile stresses, temperature and others. However, the stress intensity factor must be over a threshold value K_IH,_ and hydrogen concentration must exceed a critical value [95]. Moreover, the hydride fracture must occur close to the crack tip for the crack to propagate [26]. Following this, the temperature-dependent diffusion of mobile hydrogen is necessary. At room temperature, we observed hydrides on the cross-sections, but not the cracks (except the fracture tests) as hydrogen diffusion was too slow, and no sufficient stresses appeared. The lowest test temperatures for testing this phenomenon so far were 123 °C [95], 157 °C [96], 162 °C [14,86], 182 °C [57], and 232 °C [97]. The existence of a lower limit for the hydrogen concentration in the solution was also postulated below that where DHC could not occur [98]. It is in agreement with our results.

### 4.4. Effect of Hydrogen on Fracture Mechanism and Path

In this study, the metal matrix fractured by ductile mechanism and, almost exclusively, through the transgranular crack path. The cleavage or quasi-cleavage fracture was observed only at the hydride–matrix interface. As no mechanical stresses were imposed during homogenizing treatment, only lines of hydrides were found, and no cracks were observed, as shown in Figure 7 and Figure 9. However, during impact loading resulting in fracture, the cracking proceeded lines of hydrides directly, as displayed in Figure 8 and Figure 10. 

The results are in agreement with some previous research. Bertolino et al. [20,38,50] noted almost exclusively ductile failure and attributed such behavior to the precipitation of hydrides parallel to the rolling direction. Otherwise, locally brittle failure was noticed. Such a ductile fracture mechanism comprises a series of subsequent events: nucleation, growth, and coalescence of microcavities. However, at increasing hydrogen content, the precipitates guided the fracture path. The brittle fracture appears with an appearance of hydride precipitates and depends on their orientation [99]. The local hydrogen concentration is governed by the overall hydrogen content of the material relatively low in our experiments, resulting in a small number of hydride precipitates. Therefore, even at room temperature, the fracture proceeds according to the ductile mechanism and only locally by cleavage. The hydrogen content may explain the differences in fracture mode, from ductile in slightly charged specimens to brittle fracture in specimens heavily charged with hydrogen, as shown in Figure 10. The fracture may proceed then by either void growth and coalescence, or by hydride cleavage [100], and here the first mechanism is mainly active, and the brittle cleavage fracture occurs when the crack propagating during fracture test proceeds by the hydrides’ line. Two typical hydrogen-enhanced degradation forms are then present simultaneously: hydrogen-enhanced localized plasticity (HELP) in which hydrogen decreases the crack necessary for the crack to propagate, and hydrogen-enhanced decohesion (HEDE) observed as cleavage precisely when the crack moves through the hydride platelet. 

### 4.5. Effect of Hydrogen Influence on Mechanical Properties 

The effects of hydrogen on stress–strain curves in slow strain rate tests were minor. The only distinct loss of elongation after oxidation at 900 °C is likely to originate from grain recrystallization at this high temperature, as no hydrogen can enter the specimen at a current value of zero. The hydrogen absorption seems to result in moderate softening, likely by interstitial hydrogen, as discussed later. The results are by some earlier reports [36,37]. The effects of interstitial hydrogen and hydride phases are opposite, but the distinct loss of plasticity observed in other studies [39,40,41,42] can appear at high hydrogen content and a significant number of hydrides. The separation of these effects is not easy at low hydrogen content. The slow strain rate tests have never been made for simultaneously tensed, and hydrogen-charged Zr alloys. For many metallic materials, the slow strain rate tests in the corrosive or hydrogen-containing environment have shown the initiation of cracks on the surface and their significant progress during tensile tests. Such behavior has been considered due to the hydrogen screening of dislocations, a decrease of their mutual interaction, and movement of dislocations at lower flow stress, i.e., local softening of material at macroscopically brittle fracture known as the HELP mechanism. As shown by these results, such behavior seems impossible for “frozen” hydrogen. Locally, oversaturated interstitial hydrogen solution may decrease the dislocation interaction energy and be a cause of softening, and the HELP mechanism might operate. However, like in other metals in which hydrogen solubility is very low at room temperature, and hydride forms may appear, the HEDE can be an active mechanism at low hydrogen contents and HDC at higher temperatures and hydrogen contents. Regardless, conventional tensile tests cannot decide what mechanisms are critical for hydrogen embrittlement at these specific conditions, low hydrogen amounts and room temperature. 

The proposed model is compatible with the results of mechanical tests. In this study, hydrogen did not affect the mechanical strength within the limits of experimental error. The small increase in ductility may be attributed to the effect of the appearance of an oversaturated solid hydrogen solution. 

### 4.6. Hydrogen Influence on Nanomechanical Properties

The nanoindentation measurements were helpful in the past to recognize the mechanisms of hydrogen embrittlement for many metals. Such attempts were performed not only for Zr alloys [44,45,46,47,48,101], but also for iron single crystal [102], carbon steel [103], manganese steel [104,105], austenitic steel [106], super-duplex steel [107], nitrided steel [108], Al [109], V [110], Ni [111,112,113], Pd [113], and Ni-Nb-Zr [112], CoCrFeMn [114] and Fe-3Si [115] alloys.

The observed changes in hardness and Young’s modulus with hydrogen content are caused by the softening that may be attributed to interstitial hydrogen and by the hardening due to hydride phases. Figure 15 demonstrates the proposed runs of two components related to hydrogen in matrix and content. The shown surface hydrogen contents were calculated, taking into account the average hydrogen contents shown in Figure 6 and the known Sievert’s law, i.e., proportional relation of hydrogen content to cathodic current density. Figure 16 shows, very approximately compares total surface hydrogen content at one arbitrary applied hydrogen diffusion coefficient at room temperature. 

The observed hardening can be easily explained. At room temperature, the hydrides appear even at shallow hydrogen content. In past research, the hardness of the hydride phase was observed as a little higher than that of zirconium [46,47,48,49]. In these experiments, the hardness in highly hydrogen-charged zirconium was about 6 GPa, much higher than for the noncharged alloy, likely due to high hydrogen content and a number of hydrides within a thin subsurface layer. 

The minimum was also observed for Young’s modulus. Again, when considering the high hydrogen contents, it is in correspondence with previously reported research in which Young’s modulus of hydride has been higher than that of the Zr matrix [46,47,48]. Therefore, it cannot be ascribed to the lattice decohesion as postulated for other non-hydride-forming metals [109,111]. 

Providing a clear explanation for the mechanism resulting in the observed softening proves to be more challenging. In several tests in which the pop-in load decreased with the increase in hydrogen concentration, and multiple pop-ins were observed in the load–displacement curves, this phenomenon was explained by nucleation of dislocations by hydrogen atoms, followed by an interaction between the dissolved hydrogen atoms and the newly formed dislocation loops, and the reduction of their line energy [107,108,111,115]. However, such tests and concepts have never been proposed for zirconium or other metals in which hydrides are formed. We think that the concept of dislocation nucleation caused by mobile hydrogen may be valid for metals in which hydrogen solubility is significant at room temperature, and hydrogen diffusion is fast. However, for this specific case of zirconium highly saturated with hydrogen and investigated at room temperature, another model can be proposed. 

So far, nanoindentation tests on hydrogen-charged Zr alloys were made at relatively low hydrogen contents, and they were usually limited to the measurements on hydrogenated Zr alloy or separately on the matrix and hydride phase. The observed softening can be attributed only to hydrogen effects on dislocations. Two possible mechanisms can be considered: (i) the enhancement of nucleation of dislocations associated with local plasticity as predicted by the defactant model [116,117], or (ii) a local decrease in interaction energy of dislocations by interstitial hydrogen distributed close to or pinned to them, as postulated in the classical HELP model. It is then necessary that hydrogen is present in a lattice and pins dislocations. According to the defactant model, the softening occurs if hydrogen atoms segregate to kinks, and the formation of kink pairs is the determining step. Hardening appears at higher hydrogen contents if the rate of dislocation glide is determined by kink motion. The hydrogen segregation on the dislocation lines is then responsible for nucleation of new dislocations by reducing the line energy of the dislocation by pinned hydrogen atoms. Zhao et al. [103] supported the above model suggesting that both softening and hardening in hydrogen-charged metals could occur. They explained the softening as due to an increased mobility of screw dislocations by reducing their Peierls’ potentials or by nucleating double kinks resulting in softening at low H contents. The hardening was assumed by H segregation at dislocations followed by increasing stress required for plastic deformation. Such an explanation of nanoindentation results was proposed for several metals and alloys. 

These models could be applicable, but in our tests, the profound decrease in hardness was observed even at high hydrogen content. The possible localized plasticity was previously rejected, assuming that in annealed crystal, the number of dislocations is insufficient for their significant interaction, and subsequently, also for a potential decrease in their interaction energy [102,104,106,107,110]. The reducing lattice cohesion [108,111] was also postulated. We think, however, that the sufficient number of new dislocations is created by the nanoindenter tip within the surface layer, exactly where the oversaturated hydrogen solution is present. Moreover, if hydrogen presence on dislocation lines could result only in the generation of new dislocations, a local hardening of a matrix by the formation of dislocation pile-ups would be observed instead of softening. Thus, the nanoindentation method is a unique one that allows for in situ studies of hydrogen interaction with freshly created dislocations, and these results are evidence that hydrogen locally decreases the plasticity, likely by a mechanism postulated in the HELP model. The hydrogen-enhanced generation of dislocations may be applied to explain the hydrogen-related hardening sometimes observed, but not the softening. 

The gradual decrease in hardness after charging is due to hydrogen desorption following the decomposition of hydrides. The recovery process occurred partially as the final hardness value was about 5.5 GPa, much higher than 3.5 GPa measured for the reference specimens. That assumption regarding a substantial contribution of cracks into hardness was already expressed [102,103,104,105]. 

## 5. Conclusions

The thickness of oxide layers increasing with temperature follows kinetic laws. The oxide layers have a complex structure and are comprised of metallic elements, oxygen, and nitrogen. The descaling of the oxide layer appears at a relatively high temperature, which may be attributed to the relatively low roughness of the surface.

The barrier effect of the oxide layer can be noticed after oxidation above 500 °C, and is directly related to the increasing layer thickness. The hydrogen diffusion coefficient in the oxide phase at room temperature is between 0.6 × 10^−10^ cm^2^/s and 6 × 10^−16^ cm^2^/s. Despite the substantial degradation of the oxide layers after oxidation at 900 °C and 1000 °C, they remain effective barriers, likely due to an appearance of a two-zone structure with an inner layer more bound to the surface. 

Hydrogen entering the Zr alloy causes the appearance of hydrides. The cracking does not occur because of very slow hydrogen diffusion at room temperature and the absence of sufficient mechanical stresses. 

The Zr alloy uniformly charged with uniform hydrogen distribution at its content below or at about 100 wt. ppm reveals the ductile cracking mechanism and mainly the transgranular crack path. The cleavage or quasi-cleavage fracture appears only at the hydride–matrix interface—the cracking proceeds directly along the chains of hydrides.

The hydrogen content determines the fracture mode, more ductile in slightly hydrogen-charged specimens, and more brittle in samples heavily charged with hydrogen. Two hydrogen-enhanced degradation mechanisms are reliable at room temperature: (i) hydrogen-enhanced localized plasticity (HELP) in the matrix saturated with interstitial hydrogen and (ii) hydrogen-enhanced decohesion (HEDE) at the matrix – hydride interface. 

The effects of hydrogen on stress–strain curves in slow strain rate tests do not appear during the simultaneous tension test and hydrogen charging, and are very slight for Zr alloy that is hydrogen charged, homogenized at 400 °C and then tensed. The conventional tensile tests are improper for determining what mechanisms are critical for hydrogen embrittlement at low hydrogen amounts and room temperature. 

In nanoindentation tests, the hydrogen-related changes in hardness and Young’s modulus are caused by overlapped softening due to interstitial hydrogen and hardening due to hydrides. The softening is caused by new dislocations created by the nanoindenter tip within the surface layer, whose interaction energy is lowered by the hydrogen atoms pinned to these dislocations. The hardening is a result of the hardness of the hydride phase being higher than that of the Zr matrix. The gradual decrease in hardness after charging is due to hydrogen desorption following the decomposition of hydrides. 

## Figures and Tables

**Figure 1 materials-13-01913-f001:**
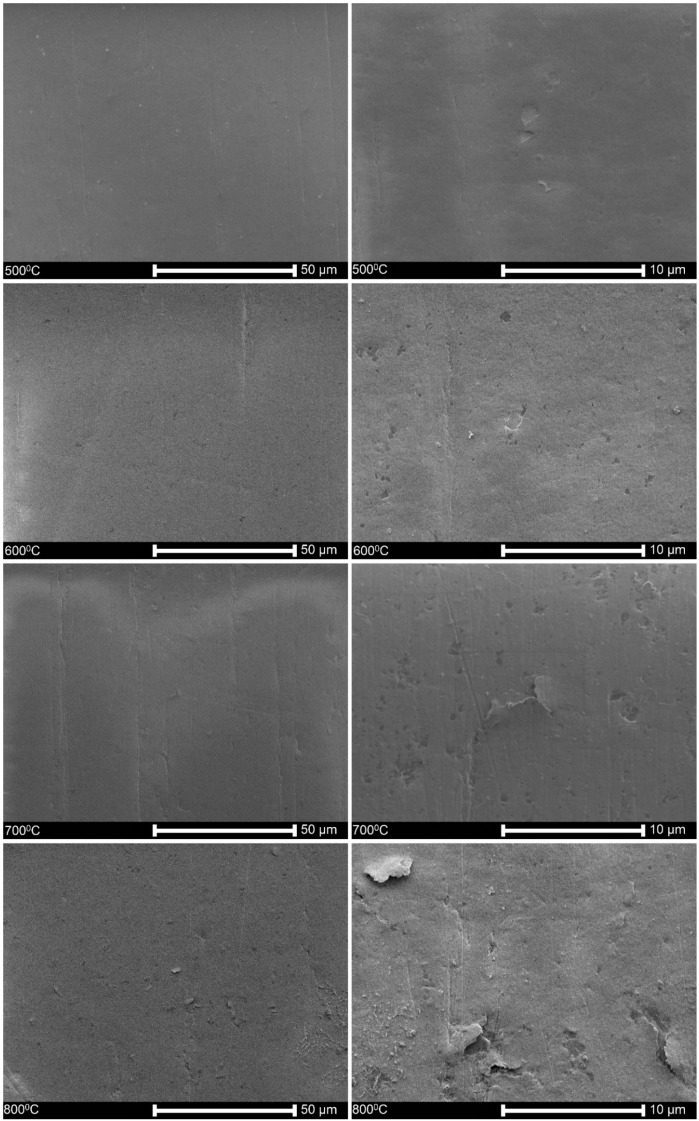
Surface topography of the alloy oxidized at different temperatures.

**Figure 2 materials-13-01913-f002:**
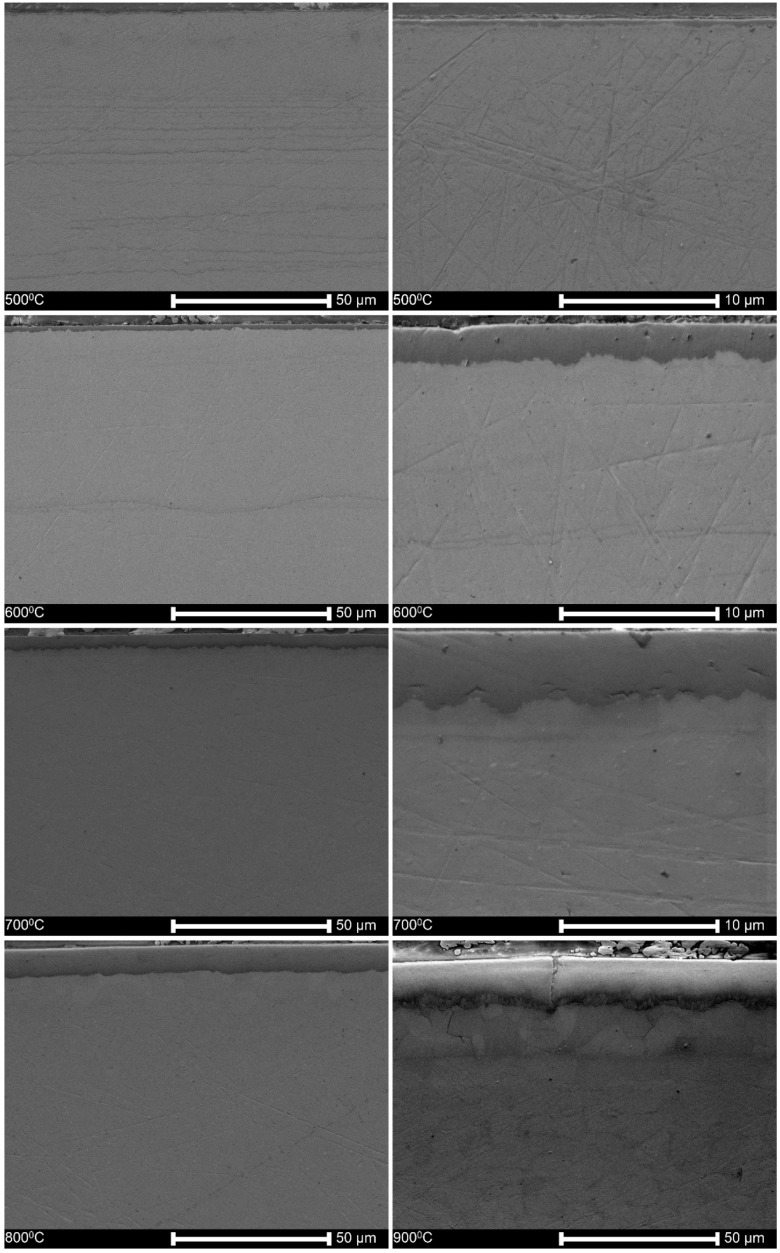
The cross-sections of the oxide layers obtained at different temperatures, between 500 °C and 1000 °C, as shown in each photo.

**Figure 3 materials-13-01913-f003:**
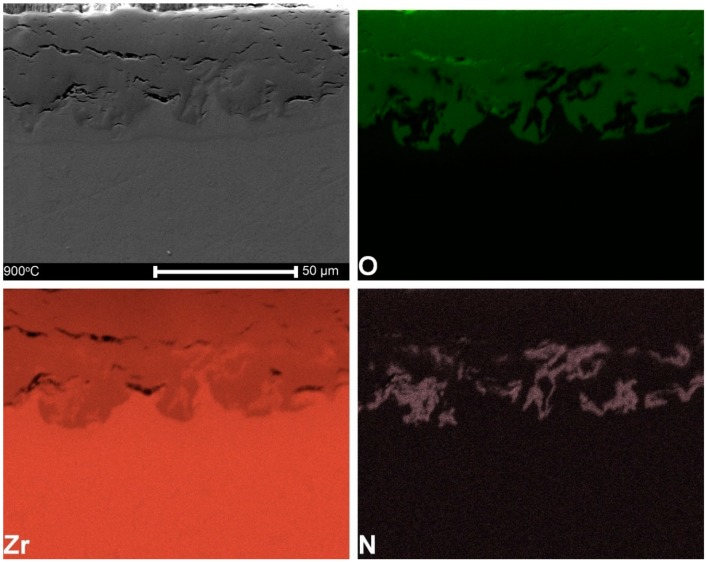
The mapping of primary elements within the oxide layer formed after oxidation at 900 °C.

**Figure 4 materials-13-01913-f004:**
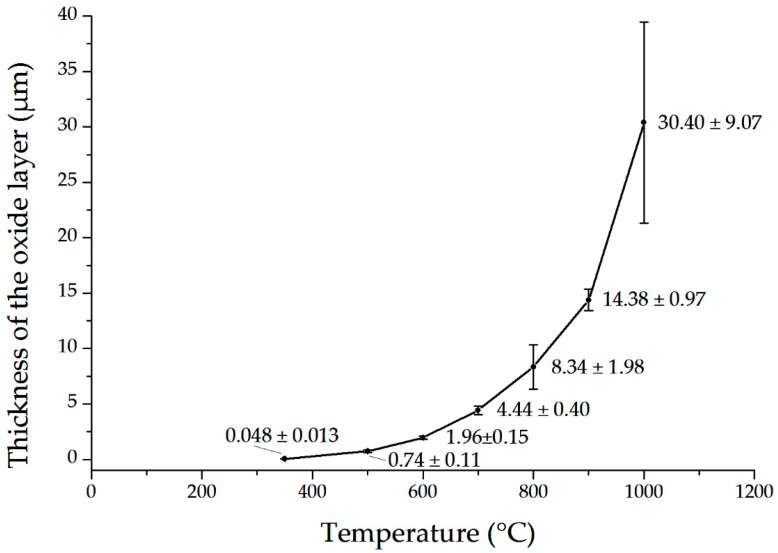
The thickness of the oxide layer formed at a different temperature for 30 min. The means and standard deviations based on five independent measurements are shown.

**Figure 5 materials-13-01913-f005:**
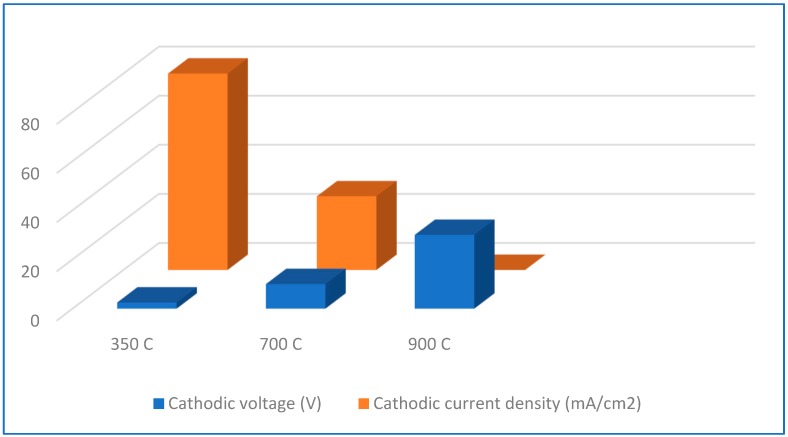
Voltage and current density values for specimens oxidized at different temperatures and subject to hydrogen charging.

**Figure 6 materials-13-01913-f006:**
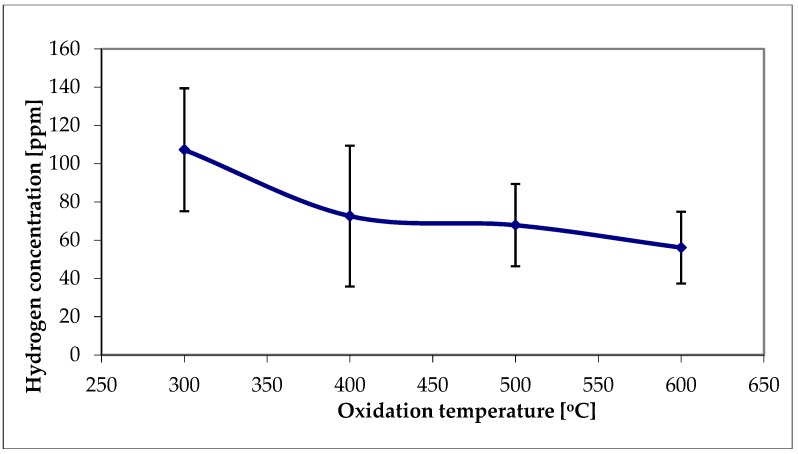
Effect of oxidation temperature on a hydrogen content. The means and standard deviations based on six independent measurements are shown.

**Figure 7 materials-13-01913-f007:**
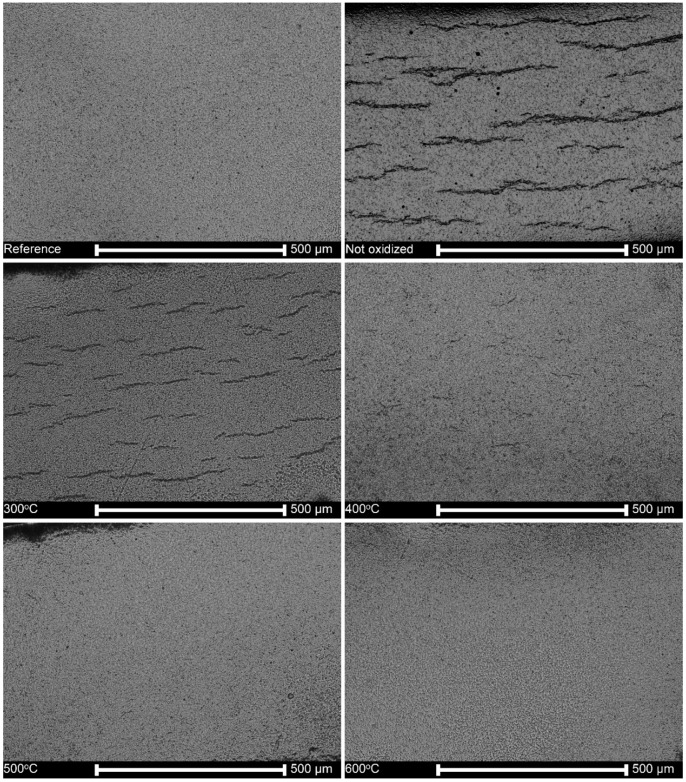
Cross-sections of specimens oxidized at different temperatures and hydrogen-charged at a constant current density of 80 mA/cm^2^ for 3 days and homogenized at 400 °C for 4 h.

**Figure 8 materials-13-01913-f008:**
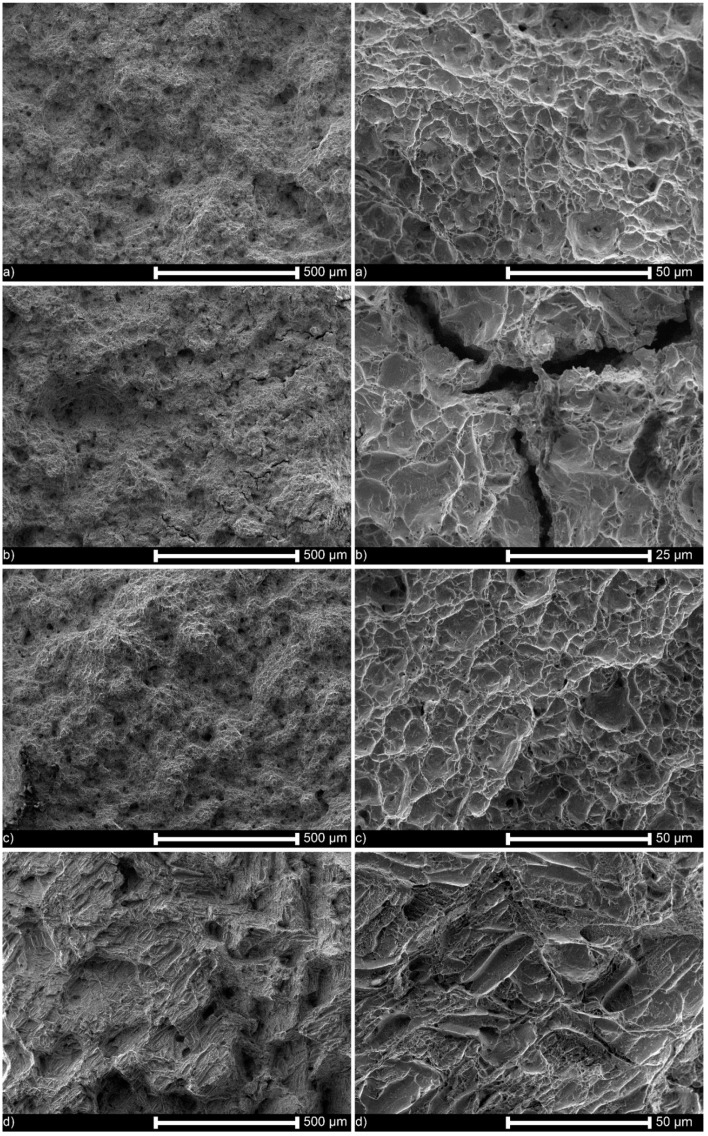
Fracture faces of specimens: (**a**) oxidized at 350 °C, and simultaneously hydrogen charged and tensed, (**b**) oxidized at 350 °C, hydrogen charged and tensed, (**c**) oxidized at 700 °C, hydrogen charged and tensed, (**d**) oxidized at 700 °C and 1000 °C, and tensed,(**e**) oxidized at 700 °C and at 1000 °C, hydrogen charged and tensed, (**f**) oxidized at 900 °C and tensed, (**g**) oxidized at 900 °C, hydrogen charged and tensed, (**h**) oxidized at 350 °C and tensed (reference specimen).

**Figure 9 materials-13-01913-f009:**
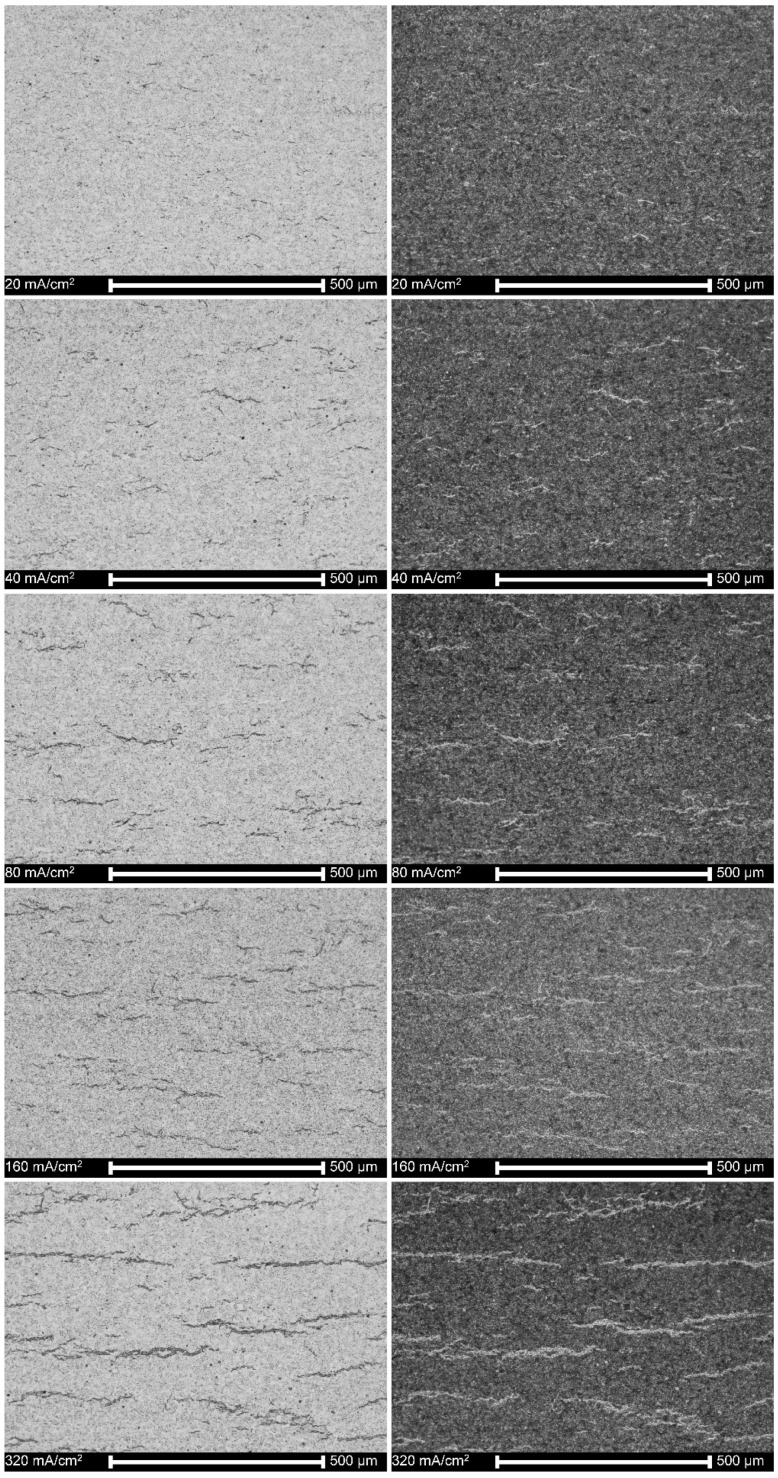
The cross-sections of nonoxidized specimens charged at different cathodic current densities (on left—the light area; on right—the dark area).

**Figure 10 materials-13-01913-f010:**
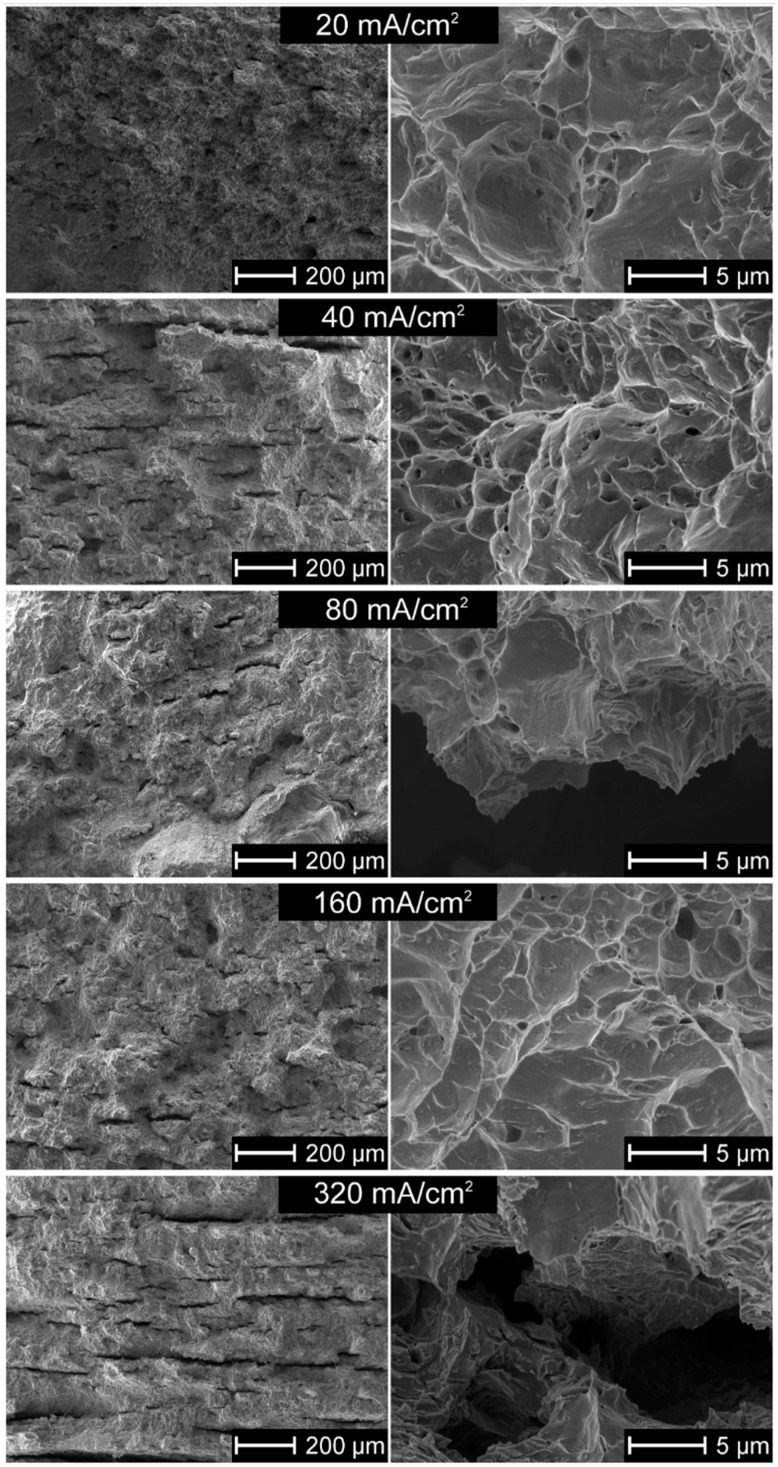
Fracture faces of nonoxidized specimens charged at different cathodic current densities.

**Figure 11 materials-13-01913-f011:**
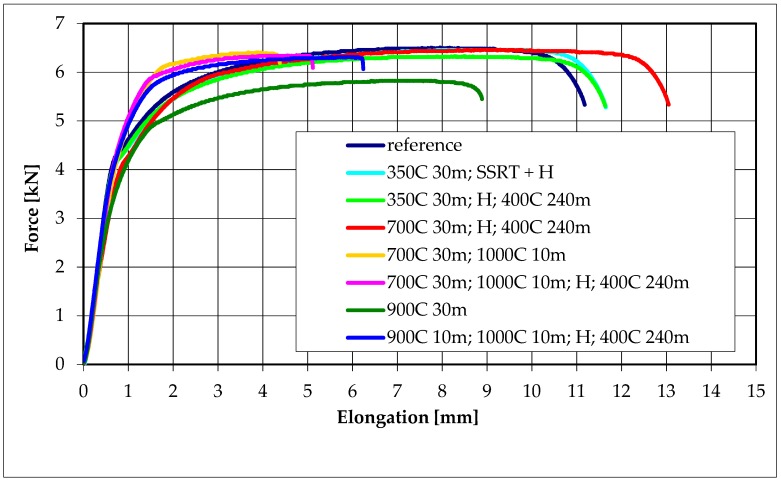
Stress–strain relationships for specimens oxidized at different temperatures of 350 °C, 700 °C and 900 °C followed by oxidation at 1000 °C for 10 min, either hydrogen charged during slow strain rate test or hydrogen charged at a constant current density of 80 mA/cm^2^, homogenized at 400 °C for 4 h and then tensed.

**Figure 12 materials-13-01913-f012:**
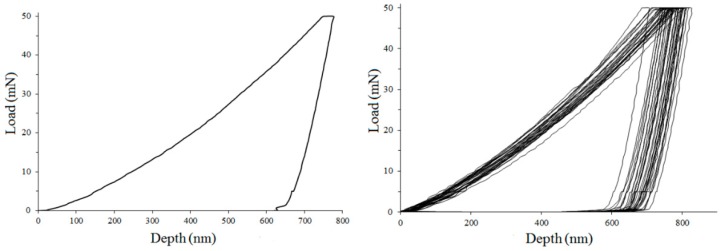
Typical stress–strain plots for nanoindentation test.

**Figure 13 materials-13-01913-f013:**
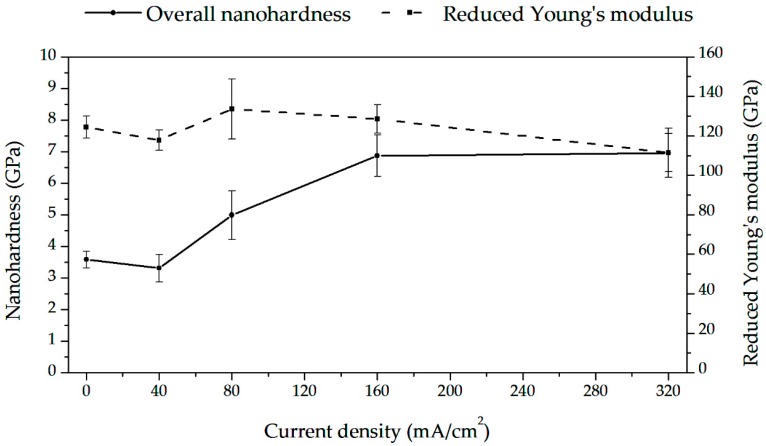
Hardness and reduced Young’s modulus vs. cathodic current density. The means and standard deviations based on ten independent measurements are shown.

**Figure 14 materials-13-01913-f014:**
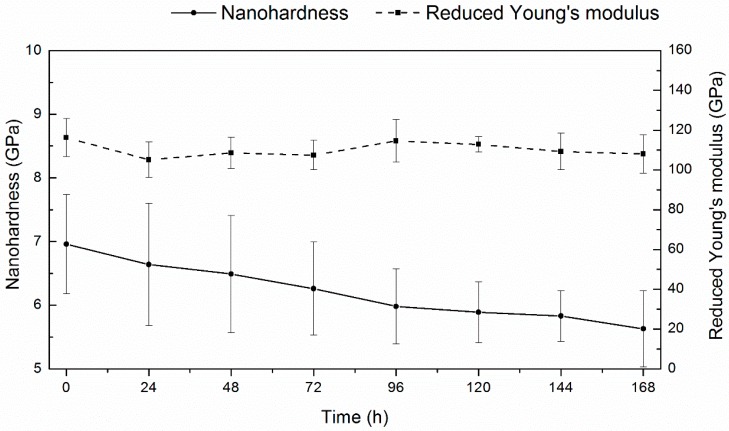
Hardness and reduced Young’s modulus of the specimen charged at 320 mA/cm^2^ and after left at room temperature for hydrogen desorption. The means and standard deviations based on ten independent measurements are shown.

**Figure 15 materials-13-01913-f015:**
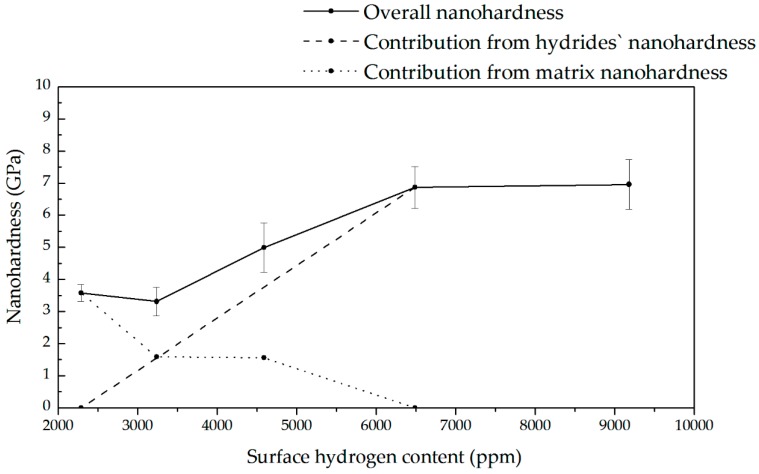
Proposed runs of two components, hydride and hydrogen interstitial solutions, of hardness over surface hydrogen content. The means and standard deviations based on ten independent measurements are shown for the upper curve.

**Figure 16 materials-13-01913-f016:**
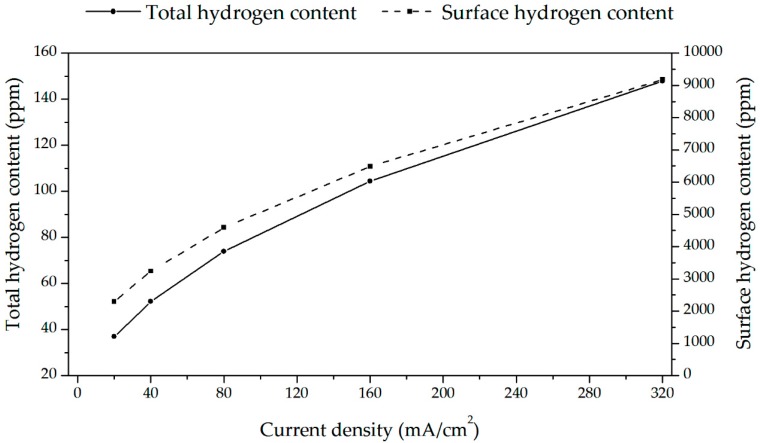
Recalculation of total hydrogen content to surface hydrogen content at different cathodic current densities.

**Table 1 materials-13-01913-t001:** Experimental plan.

Stage Number	Test Kind	Determinants	Values	Measured Variables	Characterization Technique
1	Oxidation	Oxidation temperature	300, 400, 500, 600, 700, 800, 900, 1000 °C	Surface state Oxide layer thickness	SEM
2a	Oxidation followed by hydrogen charging	Oxidation temperature Cathodic current density	350, 700, 1000 °C 80 and 300 mA/cm^2^	Tensile properties	Slow Strain Rate Test
2b		Oxidation temperature Cathodic current density	300, 400, 500, 600 °C 80 mA/cm^2^	Hydrogen content	Nondispersive Infrared Absorption
2c		Oxidation temperature Cathodic current density	300, 400, 500, 600, 700, 800, 900, 1000 °C 80 mA/cm^2^	Presence of hydrides	SEM
3	Hydrogen charging	Cathodic current density Absorption time	40, 80, 160, 320 mA/cm^2^ 72 h	Nanoindentation mechanical properties	Nanoindentation
4	Hydrogen desorption	Cathodic current density Desorption time	320 mA/cm^2^ 168 h	Nanoindentation mechanical properties	Nanoindentation

**Table 2 materials-13-01913-t002:** Nanoindentation properties measured at different current densities.

Cathodic Current Density (mA/cm^2^)	Plastic Work (nJ)	Elastic Work (nJ)	Maximum Depth (nm)
0	14.21 ± 0.67	2.59 ± 0.17	779.71 ± 27.46
40	15.08 ± 1.03	2.72 ± 0.25	816.56 ± 51.87
80	9.20 ± 1.03	3.24 ± 0.43	671.74 ± 48.09
160	9.54 ± 0.85	3.65 ± 0.24	585.95 ± 25.51
320	8.17 ± 0.44	4.00 ± 0.36	594.16 ± 35.82

**Table 3 materials-13-01913-t003:** Nanoindentation properties measured at different hydrogen desorption times for the specimen charged at 320 mA/cm^2^.

Desorption Time (h)	Plastic Work (nJ)	Elastic Work (nJ)	Maximum Depth (nm)
0	8.17 ± 0.44	4.00 ± 0.36	594.16 ± 35.82
24	8.24 ± 0.45	4.52 ± 0.10	620.08 ± 41.95
48	8.60 ± 0.49	4.32 ± 0.17	621.48 ± 44.04
72	8.60 ± 0.50	4.31 ± 0.08	630.75 ± 36.92
96	8.40 ± 0.27	4.10 ± 0.21	634.44 ± 34.87
120	8.43 ± 0.42	4.14 ± 0.09	639.22 ± 21.09
144	8.69 ± 0.68	4.19 ± 0.16	645.28 ± 26.08
168	8.69 ± 0.72	4.19 ± 0.12	657.21 ± 38.38

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
