# Peer review of "Hydrogen Embrittlement and Oxide Layer Effect in the Cathodically Charged Zircaloy-2"

_materials, 2020, doi:10.3390/ma13081913_

Round 1

Reviewer 1 Report

Even if the subject discussed in the article is interesting for the results obtained, the reading of the article was laborious and complex for me, even if the overall quality of the article is good.

In my opinion, more than an article this paper can be considered a book chapter by the number of citations and bibliographical references made everywhere.

The citations included in some cases reinforce the result, in others weaken it. But it is still an interesting and valuable scientific contribution in the deepening of the influence that hydrogen and in general oxidation can have on the mechanical characteristics of Zircalloy.

A revision of the English language is necessary because some sentences are unclear or poorly formatted.

I found the discussions to be a little too elaborate than what was then summarised in the conclusions, where all the work is done and the assessments that emerged from the discussion are summarised in a few lines that, in my opinion, do not collect what was reported in the previous pages.

In the introduction, the scientific novelty, compared to the state of the art, is well described.

In materials and methods, however, where the organisation of tests is discussed, there is no table that simplifies the reading and should also be explained better what kind of experimentation has been adopted. Also because we’re talking about two replicas used for each temperature, so I’d expect a randomized, maybe factorial plan.

Figure 1 starts from the temperature of 500°C, while in the text, to line 180, it is referred to temperatures “...from 400°C”.

The sentence at line 190 must be revised because something is missing to understand its meaning.

Figure 2 is difficult to understand, especially for the temperature of 1000°C. Perhaps you could organize differently to better understand that you are seeing the same image but with a different magnitude and so more detailed.

At line 224 the authors write: "..., as a rule,...". What does that mean? is there a norm or some citations that might help to understand this common rule?

Figure 6 is not clear at all. You have to explain the horizontal graph better, the legend is indecipherable and also the standard deviation is so high that I think you have few measured or measurable experimental samples.

The phrase at lines 257-258 should be revised because the English form is unclear.

At line 270 I don’t think it’s right to write "...permit to compare...".

at line 271 what are these "two procedures?"

Figure 8 is unclear. The caption is too long and complex and the reference letter is too small and difficult to read. I would also clarify in the text the reference to the lack of "cracks" in Figure c) 50 micron. If so, then the only image with "cracks" would be b) 25 microns?

Figure 9 appears only at the end in the discussions while it is not introduced as for the others in the results. Why?

At line 301, the form of the phrase in English should be revised because it is unclear.

At line 320 you speak about oxidation at 900°C but there is no trace in figure 11. To what you refer then?

Where is Figure 13 introduced?

At line 349 you cite "tests", but it would be advisable to put in parentheses the reference to the table suggested above in Materials and Methods.

The explanations of the terms of the equation at line 419, which should be numbered, would be better to be listed.

The sentence at lines 482-483 should be revised because the English form is unclear.

Figure 14 was used on page 16 and then on page 22. Check the Figures numbers.

In the conclusions, at lines 612-613 revise the English form. The meaning of the sentence is not clear.

Author Response

Answers to the Reviewer 1

 Reviewer: Even if the subject discussed in the article is interesting for the results obtained, the reading of the article was laborious and complex for me, even if the overall quality of the article is good. In my opinion, more than an article this paper can be considered a book chapter by the number of citations and bibliographical references made everywhere. The citations included in some cases reinforce the result, in others weaken it. But it is still an interesting and valuable scientific contribution in the deepening of the influence that hydrogen and in general oxidation can have on the mechanical characteristics of Zircalloy.

Answer: Thank you very much for your positive opinion. We have discussed this remark and decided to show the state-of-the-art as it is now, together with some contradictory results. Besides, even if the number of references is very high, they are necessary as the research focuses on two aspects: the role of the oxide layer in hydrogen degradation and the mechanism of hydrogen embrittlement related to the hydrogen content. 

Reviewer: A revision of the English language is necessary because some sentences are unclear or poorly formatted.

Answer: A native English speaker has already reviewed the manuscript, and now we have also checked it with the Grammarly Premium deleting some still present errors. Besides, we have changed all sentences, which could be unclear.

Reviewer: I found the discussions to be a little too elaborate than what was then summarised in the conclusions, where all the work is done and the assessments that emerged from the discussion are summarised in a few lines that, in my opinion, do not collect what was reported in the previous pages.

Answer: Thank you for your fundamental remark. Following that, we have enriched and improved the discussion.

Reviewer: In the introduction, the scientific novelty, compared to the state of the art, is well described.

Answer: Thank you for your kind statement.

Reviewer: In materials and methods, however, where the organisation of tests is discussed, there is no table that simplifies the reading and should also be explained better what kind of experimentation has been adopted.

Answer: Thank you for this essential remark. Following that, we have added the table showing the experimental plan and used methods and explained its design aims.

Reviewer: Also because we’re talking about two replicas used for each temperature, so I’d expect a randomized, maybe factorial plan.

Answer:: In our experiment, we have applied a plan, which regretfully has not been clearly described. In this specific case, a factorial design could not be used as we have not planned to test the influence of some process parameters on the output variables. Besides, writing “…two replica” we have made an error, thinking of one specimen and its two replicates so that three specimens each time. The proper description is in the text below and the corrected manuscript. Now, the new part is introduced as follows:

The experimental plan was comprised of several stages as shown in Table 1. In the first stage, the effect of oxidation temperature on the oxide layer thickness and degradation was analyzed, establishing in such a way the proper oxidation temperatures for the next tests. Afterward, the slow strain rate tests, together with or after hydrogen loading, were performed to select the procedure for an assessment of mechanical properties (No. 2a). For some specimens, oxidized at high temperature, the cathodic charging at 300 mA/cm2 was also done made to know whether, for highly oxidized specimens, the increase in current density could result in any effects in slow strain rate tests. In the next steps, the effects of the oxide layer thickness on the hydrogen content (No. 2b) and presence of hydrides in the bulk (No. 2c), were carried out determining in such a way the barrier function of the oxide layer. In the final steps, made on no- oxidized specimens, the effects of hydrogen amount, determined by cathodic current density (No. 3), and hydrogen desorption, determined by desorption time (No. 4), on nanoindentation mechanical properties, were examined. All tests were performed on at least three specimens, randomly selected among all prepared at the beginning of a research.

Reviewer: Figure 1 starts from the temperature of 500°C, while in the text, to line 180, it is referred to temperatures “...from 400°C”.

Answer:  The appropriate change has been made.

Reviewer: The sentence at line 190 must be revised because something is missing to understand its meaning.

Answer:  The appropriate change has been made.

Reviewer: Figure 2 is difficult to understand, especially for the temperature of 1000°C. Perhaps you could organize differently to better understand that you are seeing the same image but with a different magnitude and so more detailed.

Answer:  The figure has been described more in detail in the manuscript. We have decided not to show all pictures in the same image assuming that some details may be lost.

Reviewer: At line 224 the authors write: "..., as a rule,...". What does that mean? is there a norm or some citations that might help to understand this common rule?

Answer:  The appropriate change has been made. We have applied the current density 80 mA/cm2 for all specimens. Still, some experiments were also performed to know whether, for highly oxidized specimens, the increase in current density could result in any effects in slow strain rate tests.

Reviewer: Figure 6 is not clear at all. You have to explain the horizontal graph better, the legend is indecipherable and also the standard deviation is so high that I think you have few measured or measurable experimental samples.

Answer:  The figure has been drawn again.

Reviewer: The phrase at lines 257-258 should be revised because the English form is unclear.

Answer:  Regretfully, this whole part has not been related to these results, and the necessary change has been made.

Reviewer: At line 270 I don’t think it’s right to write "...permit to compare...".

Answer:  The appropriate change has been made.

at line 271 what are these "two procedures?"

Answer:  The appropriate change has been made. In particular, the applied procedures have been described in the text, and those have been explained in the main text and not only in the caption.

Reviewer: Figure 8 is unclear. The caption is too long and complex and the reference letter is too small and difficult to read. I would also clarify in the text the reference to the lack of "cracks" in Figure c) 50 micron. If so, then the only image with "cracks" would be b) 25 microns?

Answer:  The appropriate change has been made. In particular, the reference letters have been magnified.

Reviewer: Figure 9 appears only at the end in the discussions while it is not introduced as for the others in the results. Why?

Answer:  The appropriate change has been made. In particular, the figure has been discussed also in the Results section.

Reviewer: At line 301, the form of the phrase in English should be revised because it is unclear.

Answer:  The appropriate change has been made.

Reviewer: At line 320 you speak about oxidation at 900°C but there is no trace in figure 11. To what you refer then?

Answer:  The appropriate curve has been added.

Reviewer: Where is Figure 13 introduced?

Answer:  The appropriate change has been made, it has been an error.

Reviewer: At line 349 you cite "tests", but it would be advisable to put in parentheses the reference to the table suggested above in Materials and Methods.

Answer:  The appropriate change has been made.

Reviewer: The explanations of the terms of the equation at line 419, which should be numbered, would be better to be listed.

Answer:  The appropriate change has been made.

Reviewer: The sentence at lines 482-483 should be revised because the English form is unclear.

Answer:  The sentence has been removed.

Reviewer: Figure 14 was used on page 16 and then on page 22. Check the Figures numbers.

Answer:  The numbers of figures has been changed.

Reviewer: In the conclusions, at lines 612-613 revise the English form. The meaning of the sentence is not clear.

Answer:  The appropriate change has been made.

Reviewer 2 Report

The manuscript submitted by Gajowiec et al. reports the effects of oxide layers on the hydrogen uptake and tensile strength of Zircaloy-2 alloy, as well as the interplay between hydrogen contents and mechanical properties of Zircaloy-2 alloy.

This data and discussion in this manuscript are thorough. The authors have displayed a comprehensive understanding of the oxidation, hydrogen-uptake, and changes in mechanical properties of Zircaloy-2. Nevertheless, the language of this manuscript needs extensive refining, as some sentences are awkwardly structured that impedes understanding. Additionally, there are the following issues that need clarification and elaboration:

  1. Lines 68-71: Please explain the temperature dependence of hydrogen solubility.
  2. Line 82: The meaning of "t" is unclear.
  3. In Section 2 (Materials and Methods), please specify under what atmospheres that all the annealing was operated.
  4. Line 142: What does "mixed" mean?
  5. Fig. 1: Please label clearly the temperatures for each panel. The temperatures residing at the lower left corners are small and easy to be overlooked.
  6. Fig. 4: Please explain how to determine the thicknesses, as I assume the oxide layers are non-uniform.
  7. Lines 23-234: The claim "neither dissolution nor damage of oxide layers was observed at any test" lacks evidence.
  8. Fig. 6: The name of the y-axis partially overlaps with the axis numbers. Additionally, please explain what the error bars stand for in the figure caption.
  9. Fig. 8: Please enlarge the panel numbers to make them legible.
  10. Fig. 9 is not mentioned in the text.

Author Response

Answers to the Reviewer 2

Reviewer: The manuscript submitted by Gajowiec et al. reports the effects of oxide layers on the hydrogen uptake and tensile strength of Zircaloy-2 alloy, as well as the interplay between hydrogen contents and mechanical properties of Zircaloy-2 alloy.

This data and discussion in this manuscript are thorough. The authors have displayed a comprehensive understanding of the oxidation, hydrogen-uptake, and changes in mechanical properties of Zircaloy-2.

Nevertheless, the language of this manuscript needs extensive refining, as some sentences are awkwardly structured that impedes understanding.

Answer: Thank you very much for your positive opinion. We have thoroughly reviewed the manuscript again and refined,  again checking all sentences and improving the the language with the Grammarly Premium  software.

Reviewer: Additionally, there are the following issues that need clarification and elaboration:

  1. Lines 68-71: Please explain the temperature dependence of hydrogen solubility.

Answer: The whole part has been rewritten following this suggestion.

  1. Line 82: The meaning of "t" is unclear.

Answer:  The explanation of “t”, hydrogen uptake time, has been added.

  1. In Section 2 (Materials and Methods), please specify under what atmospheres that all the annealing was operated.

Answer:  The appropriate change has been made.

  1. Line 142: What does "mixed" mean?

Answer:  The appropriate change has been made. It has been an error, it should be “stirred”.

  1. 1: Please label clearly the temperatures for each panel. The temperatures residing at the lower left corners are small and easy to be overlooked.

Answer:  The designations have been magnified.

  1. 4: Please explain how to determine the thicknesses, as I assume the oxide layers are non-uniform.

Answer:  The detailed description has been introduced in the text. It is as follows:

The thickness of oxide layers was calculated for each specimen as a mean value of five independent measurements, made at an equal distance, close to the oxide thickness.       

 Lines 23-234: The claim "neither dissolution nor damage of oxide layers was observed at any test" lacks evidence.

Answer:  It has been only our observation. We cannot give any evidence that damage has not been observed, there is no such manner, simply we compared surface s and did not observe any changes. But, now we have improved the text following this remark.

  1. 6: The name of the y-axis partially overlaps with the axis numbers. Additionally, please explain what the error bars stand for in the figure caption.

Answer:  The figure has been drawn again.

  1. 8: Please enlarge the panel numbers to make them legible.

Answer:  The designations have been magnified.

  1. 9 is not mentioned in the text.

Answer:  The appropriate change has been made.

Round 2

Reviewer 2 Report

The authors have adequately addressed most of my comments, except one request mentioned in my Comment 8: "Please explain what the error bars stand for in the figure caption."

Wherever error bars are presented, a note in the corresponding figure caption is needed to explain how the error bars are obtained. For example, are they standard deviations? Based on how many independent measurements?

Author Response

Wherever error bars are presented, a note in the corresponding figure caption is needed to explain how the error bars are obtained. For example, are they standard deviations? Based on how many independent measurements?

Answer: Following this remark, the captions of the Figures 4, 6, 13, 14, 15 and Tables 2 and 3 were supplemented with such information.